# Geomorphological Processes and Environmental Interpretation at Espalmador islet (Western Mediterranean)

**Laura del Valle Villalonga [1,2,\*], Alida Timar-Gabor [1,3] and Joan J. Fornós [2]** 

[1] Faculty of Environmental Sciencies and Engineering, Babeş-Bolyai University, 400084 Cluj-Napoca, Romania; alida.timar@ubbcluj.ro

[2] Grup de Ciències de la Terra, Universitat de les IllesBalears, 07122 Palma, Spain; joan.fornos@uib.cat

[3] Interdiscuplinary Research Institute on Bio-Nano-Science of Babeş-Bolyai University, 400084 Cluj-Napoca, Romania

[\*] Correspondence: lauradelvalle.geo@gmail.com; Tel.: +34-971173447

**Abstract:** This study presents a sedimentological and stratigraphical description of the Pleistocene deposits cropping out in Espalmador islet (Illes Pitiüses). Four major sedimentary facies including the succession of aeolian, marine, colluvial and edaphic environments are described. The sedimentological and stratigraphical analysis of these deposits allows the reconstruction of the coastal Pleistocene environmental and geomorphological history of the Espalmador islet. The coastal relief and the fluctuations of the sea level mainly control the Pleistocene coastal landscape evolution on Espalmador. Episodes of aeolian activity and dune formation related to a predominant northwestern wind direction can be linked to periods of low sea level where a high amount of marine sediment is exposed on the shelf platform.

**Keywords:** Coastal aeolianites; Pleistocene; Espalmador islet; geomorphological process

## 1. Introduction

Since the first half of the nineteenth century, the Quaternary is considered the "ice age"—fundamentally due to the existence of numerous periods of glaciation, when ice sheets many kilometers thick have covered vast areas of the continents in temperate areas. During and between these glacial periods, rapid changes in climate and sea level had occurred, and environments worldwide have been altered. These variations in turn have driven rapid changes in life forms, both flora and fauna [1]. The accumulations of coastal sedimentary successions characterize this climate variation. Therefore, the stratigraphic record allows an approximation to the reconstruction of the history of the Earth during the Plio-Quaternary, from multiple perspectives and indicators (sedimentology, paleontology, isotopes, etc.) and the study of the palaeoenvironmental evolution and the geomorphological processes that occurred [2]. Quaternary sedimentary successions are characterized by the alternation of shallow marine deposits (i.e., beaches), aeolianites, colluvial and alluvial deposits or the formation of palaeosols [3–24]. These deposits form a complex stratigraphic architecture that reflects the alternation of glacial-interglacial cycles, that can be correlated with the highstands and lowstands of the sea level eustatic curve, on a regional as a global scale [15,16,25–29]. These successions are fundamental to understand and predict the environmental changes that will affect mainly the present-day coastal environments in the near future.

Several studies document the Pleistocene and Holocene deposits along the Western Mediterranean (e.g., [15–18]), and specifically in the Balearic archipelago [5,19,23,30]. From that we can infer that the Balearic Islands constitute an excellent natural laboratory for the study of Pleistocene coastal deposits

due to the wealth, quality, the great variety, the good conservation and its continuity along the coasts of these deposits [2].

In this study, we document the sedimentary characteristics and timing of formation of the dune system cropping out along the Espalmador—an islet of the Pitiüsesis archipelago considered tectonically stable since the Pliocene [31,32] by means of sedimentology and stratigraphic analysis. Previous studies from the Espalmador islet [33–35] consist of very general descriptions. We discuss the factors that enhanced dune formation and how different and coastal processes participated in shaping the coastal dune field at the Espalmador islet in the general frame of the Pleistocene eustatic sea level changes.

## 2. Geological Setting

The small archipelago of the Illes Pitiüsesis composed of two large islands; Eivissa and Formentera, and sixty islets. They form the emerged part of the southern block of the Balearic promontory located in the westernmost part of the Mediterranean (Figure 1A,B) [30].

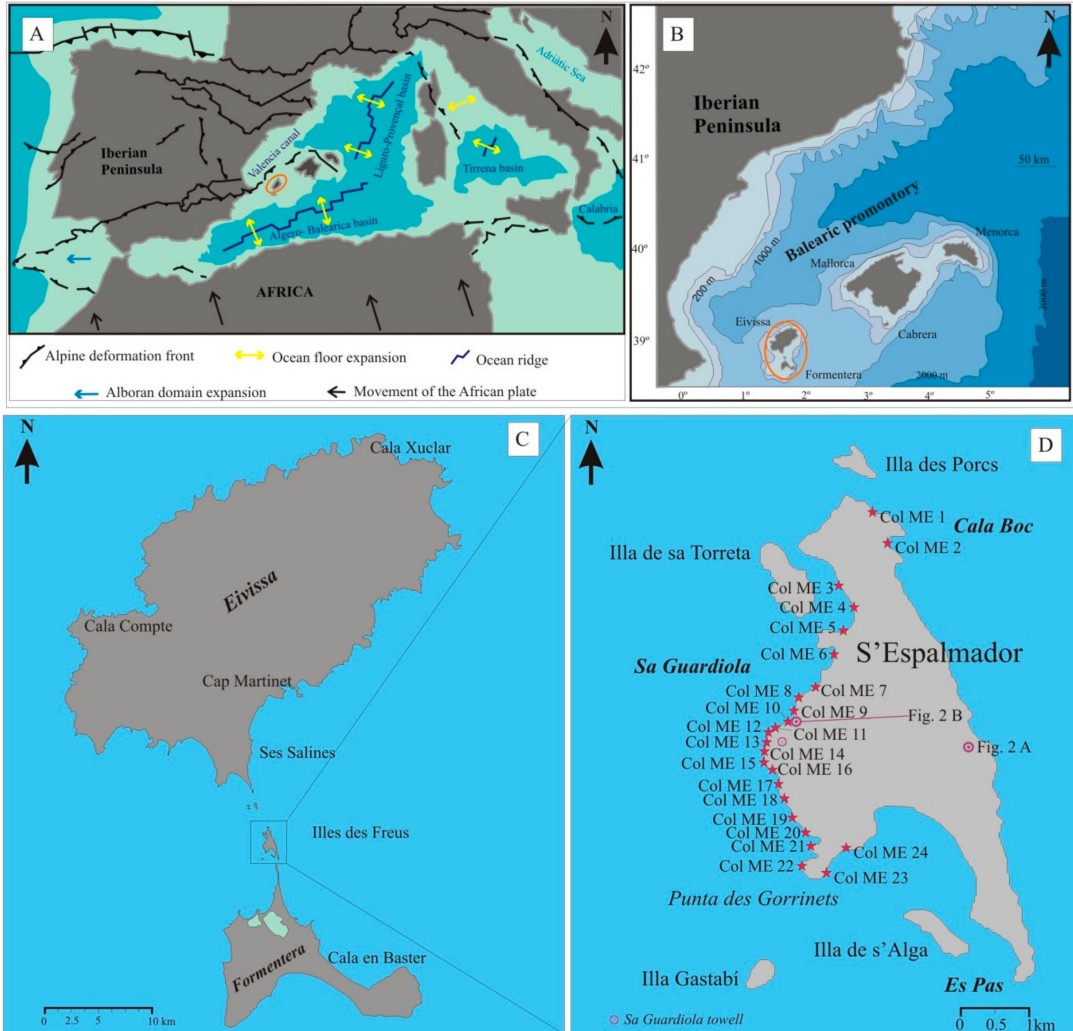

**Figure 1.** Location of the study area (**A**,**B**), the Pitiüses islands and Espalmador (**C**), and the investigated stratigraphic profiles (**D**).

Espalmador is separated from Formentera by a very narrow and shallow passage (50-m wide and 2-m deep) from the Pas cape to the es Borronar cape. The islet is surrounded by other smaller islets (es Porcs, saTorreta, Casteví and s'Alga).

The islet has an elongated shape with a north–south orientation and an area of 2 km² (2.925m long and 800m wide). The eastern slope is a coast with sandy beaches in the south and rocky formations with a small pocket beach, cala de Bocs, in the north. The northern most point is the es Faro den Pou located on the es Porcs islet and marks the boundary between Eivissa and Formentera. The western slope shows greater variability with two bays, sa Torreta and s'Alga, separated by the cliffs of the sa Guardiola cape and the main massif of the island with 22 m of height. The islet is composed entirely of Quaternary dune fields, silt-sandy palaeosol with calcrete crusts, partially covered by Holocene dunes (Figure 2A). The basement of the island is formed by conglomerates appertaining to the Miocene, being the continuation of Formentera's basement (Figure 2B). The current morphology of the islet coast has been gradually destroyed by the abrasion of post-Miocene marine transgressions [35].

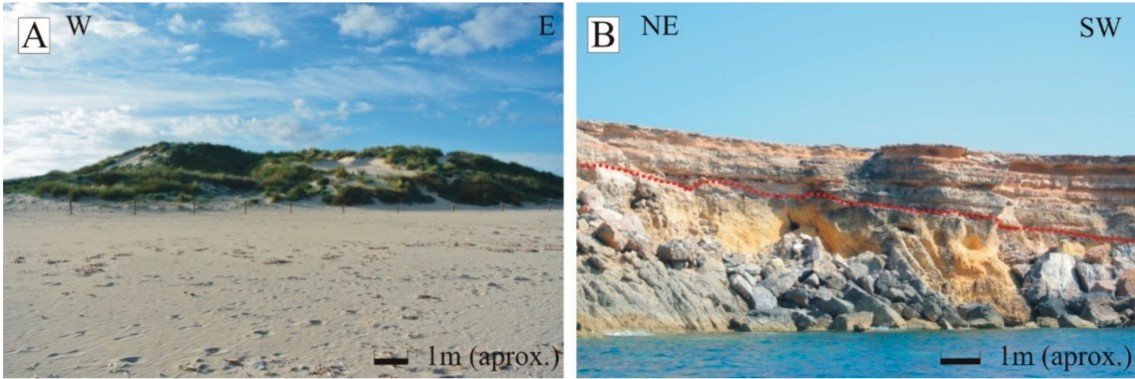

**Figure 2.** (**A**) Holocene dune fields from the Espalmador islet. (**B**) Panoramic view of the bedrock and the Pleistocene deposits from the northwestern part of the Espalmador islet.

Nolan (1895) [33] provides one of the first Quaternary descriptions of the islet, indicating the existence of beaches bearing *Strombus* (*Persistrombus coronatus*).

The Pleistocene stratigraphy of the island was subsequently studied by [34] Butzer and Cuerda (1962) who presented an interesting stratigraphic series belonging to the end the Middle Pleistocene.

## 3. Methods

### Facies Analysis

The conventional method of lithostratigraphic logging was used [36]. Additional information on cross-bed dip direction for palaeocurrent analyses was acquired, and grain size and mineralogy analyses were carried out. The nomenclature for sedimentary facies was based on [15,16]. This nomenclature links lithology, grain size, sedimentary structures and macrofossil characteristics. Facies were named according to the main lithologic features (C: conglomerate, B: breccia; S: sandstone and P: palaeosol), prevailing texture (a: sand, m: mud, s: silt), grain size (c: cobble, d: pebbles, e: medium to very coarse sand; h: fine to medium sand), sedimentary structures (l: laminated, p: planar cross-bedded, t: through cross-bedded, u: low angle cross-bedded, g: sorting, n: massive), biogenic features (f: highly fossiliferous, r: root-casts).

Twenty-four stratigraphic columns (Figure 3) were measured and correlated based on major unconformities and homogeneous units, bounding surfaces or according to the presence of continuous palaeosols. Major changes of facies were taken into account. At each log, major units were characterized in terms of grain size, composition and mineralogy, discordance; as well as its arrangement or architecture, facies associations (geometry and sedimentary bodies).

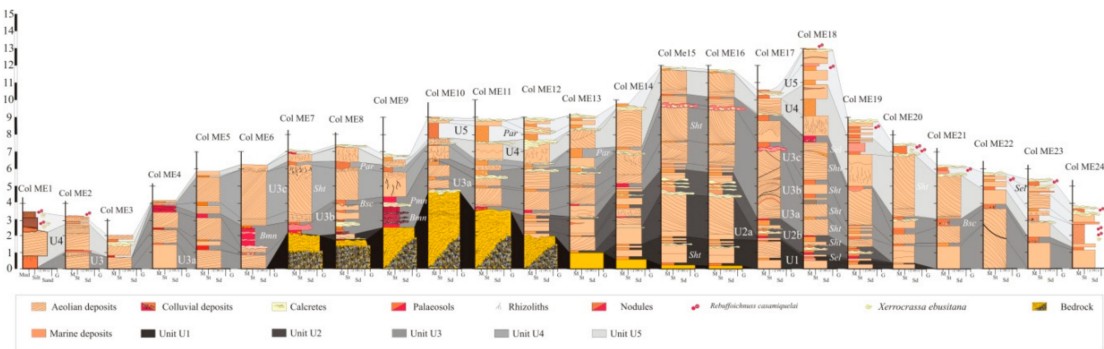

**Figure 3.** Stratigraphic columns of the sediments exposed at Espalmador (esFreus Islets). For paleowind directions see Figures 6 and 7.

A total of 21 samples of approximately 1 kg of material were collected for their textural and mineralogical characterization of the sediments. The collection of samples was done following the guidelines of Tucker, 1988 [36]. Fossil macrofauna was also collected, both marine and continental, of all levels that were possible, for its taxonomic identification. All samples showed a high degree of cementation. They were cut (0.5 × 0.5 mm) and polished to perform grain size analyses.

Images on polished faces were obtained using an optical microscope (NATIONAL Digital Microscope, Texas, USA), together with the binocular microscope with MOTIC Image ver. 2.0 software, Hong Kong, and the analysis of grain size measurements was performed using the free image analysis software IMAGEJ, contributors worldwide, Wisconsin, USA, allowing the measurement of grain size parameters (sampling of 20 grains per photo and calculation of the average of the major axis) and the determination of the composition (categories: bioclasts, lithoclasts or others).

For the remaining analyses, samples were crushed and grinded. Mineralogical content was determined using X-ray powder diffraction with a Siemens D-5000 X-ray diffractometer, Siemens Aktiengesellscft, Germany, EU using Cu Kα radiation by means of randomly oriented powders of the bulk samples of sediments. Diffraction patterns were recorded from 3° to 65° 2θ in steps of 0.03°, 0.3 s counting time per step, at 25 °C. Phase determination and semi-quantitative analysis were made by the X-Powder v.2010.01.09 Pro software using the DifData database [37]. Carbonate content has been obtained by hydrochloric acid etching [38]. Of the excess of the samples, 20 g were used to calculate the percentage of carbonates by calcimetry using a Bernard calcimeter at room temperature and comparing it with the results of the semiquantitative analyzes carried out by the XRD.

The determination of the colors was carried out by comparing a sample of sediments with the Munsell Color Table® Soil Color Chart (2000) [39] and assigning the most similar reference code.

## 4. Results and Discussion

### 4.1. Sedimentary Facies and Palaeosol Description

Sedimentology and stratigraphic analyses performed have allowed identifying four major sedimentary facies and two different palaeosols (Figures 4 and 5). Facies have been grouped in two main associations representing aeolian and colluvial sedimentary environments.

### 4.2. Aeolian Facies Association

Facies *Sht* is characterized by very pale brown (HUE 10 YR 8/2), well-sorted fine to medium-grained bioclastic sand (125–250 μm), with large-scale trough cross-stratified beds that laterally decrease in angle from 30° to 20°. Beds are 1- to 3m thick and are partially disrupted by vertical root casts (1–7 cm width and 0.5–1 m length) (Figure 3 Col ME 7). Grain composition is mainly carbonate (93%) composed of marine bioclasts with very little terrigenous material. According to X-Ray diffraction analysis, calcite is the main mineral (77%) followed by dolomite (16%). This deposit is interpreted as parabolic dunes.

| FACIES | | DESCRIPTION | INTERPRETATION | Photography |
|---|---|---|---|---|
| Aeolian facies | **Sel** | Very pale brown bioclastic sandstone. Well sorted medium to coarse-grained sand. Horitzontal to sub-horizontal bedding up to 0.5 m thick with internal lamination of 5 cm. Root traces. | **Aeolian sand-sheets** | 5 cm |
| | **Sht** | Very pale brown bioclastic sandstone. Well-sorted fine-grained sand. Trough cross-bedded. Root traces. | **Parabolic dunes** | 15 cm |
| | **Shu** | Very pale brown bioclastic sandstone. Well sorted fine to medium grained sand. Cross to horizontal bedding. Root traces. | **Parabolic dunes** | 8 cm |
| Colluvial f. | **Bmn** | Reddish muddy matrix-supported to clast-supported breccia. Clasts are angular to subangular pebbles. Channel-like structures. | **Slope-wash** | cm |
| Marine f. | **Sef** | Reddish muddy matrix-supported to clast-supported breccia. Clasts are angular to subangular pebbles. Channel-like structures. | **Beach** | 1 cm |
| Palaeosols | **Pmn** | Reddish yellow mudstone. Very plastic fine sediments banded by rich iron-layers. | | 10 cm |
| | **Par** | Very pale brown to pink silts, with sandy levels, roots casts, terrestrial fauna and nodular forms; with insect trace fossils. | | 10 cm |

**Figure 4.** Characteristics of the identified sedimentary facies and palaeosols. Their interpretation is based on main fabric features and sedimentary structures.

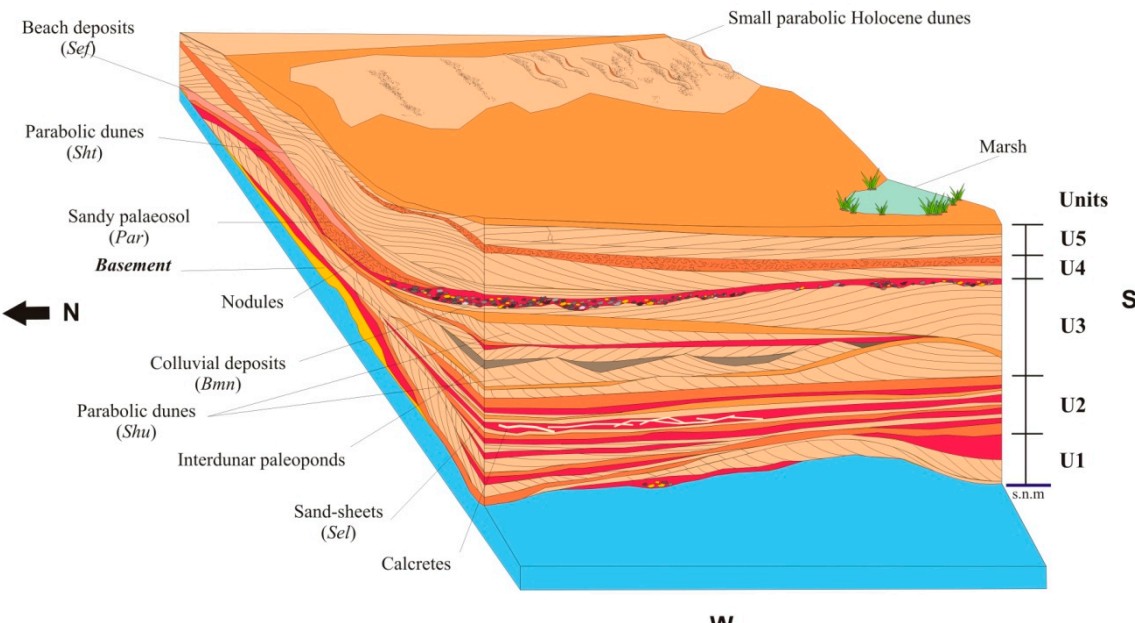

**Figure 5.** 3D landscape reconstruction of the aeolian accumulation at Espalmador islet.

Facies *Sel* is composed of very pale brown (HUE 10 YR 8/2), well-sorted medium to coarse-grained bioclastic sand (250–500 μm) and horizontal to sub-horizontal laminated structures showing a dipping angle up to 10° (Figure 3 Col ME 18). Layers are 1 to 1.2 m thick. This facies show evidences of bioturbation by root casts lightly cemented by calcite. Grains are composed of marine carbonate particles (90%) and minor amounts of quartz and feldspars (~7%). This deposit represents the aeolian transport of marine carbonate sands towards a coastal cliff and deposited as a sand sheet deposit [24].

Facies *Shu* is characterized by pinkish white (HUE 5YR 8/2) well-sorted fine to medium-grained bioclastic sand (250 μm) locally disrupted by root casts, with low-angle cross bedding (Figure 3 Col ME 18). Beds are between 1 to 1.5 m thick with internal laminations of 0.5–1 cm thick, slightly cemented by sparry calcite. Their composition is mainly carbonate (~85%) as they are made up of abundant marine bioclasts. This deposit is interpreted as small-scale parabolic dunes.

### 4.3. Colluvial-Alluvial Facies Association

Facies *Bmn* is characterized by reddish (HUE 2,5 YR 7/4) silty matrix-supported breccia with angular and heterometric clasts forming layers of millimeter to centimeter thick deposits, which are disrupted by calcretes levels (Figure 3 Col ME 9). Some breccia layers change laterally into lens-shaped structures. The abundant subangular clasts originate from the reworked upper surface of the lower level. Locally, iron bands can be observed. The mineralogical composition of the matrix is dominated by silicates (60%). This facies is interpreted as hillslope deposits.

### 4.4. Shallow-Marine Facies Association

Facies *Sef* is characterized by massive strata (maximum thickness around 40 cm) with subrounded clasts and interstices filled by medium to coarse-grained sand rich in bioclasts and marine fossils (Figure 3 Col ME 2). The colour is reddish yellow (5 YR 6/8). The following fossils were identified: *Arca noae* (Linné), *Glycymeris violascescens* (Lamarck), *Spondylus gaederopus* (Linné), *Loripes lacteus* (Linné), *Littorina neritoides* (Linné), *Littorina punctata* (Gmelin), *Thais haemastoma* (Linné), *Cerastotomae rinaceum* (Linné), *Columbella rustica* (Linné) and *Hinia costulata* (Renieri) [35]. This is interpreted as sandy beach deposits (lower shoreface).

### 4.5. Palaeosols

Two types of palaeosols are observed. The first one is characterized by massive silty-sandy texture with highly cemented of 0.5 to 2 cm thick nodules and a high degree of bioturbation by vegetation. Its colour is very pale brown (HUE 10 YR 8/4) changing locally to pinkish (HUE 7.5 YR 5/4). It bears abundant terrestrial fauna (e.g., the endemic snail *Xerrocrassa ebusitana,* and insect trace fossils *Rebuffoichnuss casamiquelai*; [40,41]). Some isolated angular clasts of millimeter to centimeter size can be observed floating in the silt-sandy material. This sandy palaeosol shows variable thickness, changing laterally from 0.5 m to 1.5 m (Figure 3 Col ME 11). In general, in the upper part 5-cm-thick calcretes are observed capping the palaeosol. The composition of the silts and sands is mainly carbonate (~80%) with calcite as major mineral (57%), followed by the aragonite (23%) and minor amounts of quartz (2%).

A second type of palaeosols are characterized by clays and silts (reddish yellow –HUE 7.5 YR 6/6 –changing to pinkish –HUE 7.5 YR 5/4–) and iron bands (Figure 3 Col ME 9). Mineralogical content shows 56% clay minerals (of which illite minerals makes up 40%), 20% quartz and a minor content of calcite (11%). Calcretes have been observed locally with pisoliths horizons and abundant vegetal traces. These consist of calcified branched filamentous structures. The red mudstone palaeosol levels show variable thickness, changing laterally from 0.10 m to 0.5 m. On the neighbouring island of Mallorca similar palaeosols have been studied, and interpreted as reflecting periods of warmer temperatures and variable conditions of aridity and vegetation cover [42,43] related to periods of sea level high-stands [30,43].

## 5. Discussion

Carbonated aeolian deposits interbedded with marine deposits, colluvial deposits and palaeosols are characteristic for the coastal Pleistocene sediments of the Espalmador islet. The morphology, consisting of small cliffs on folded Miocene conglomerates control the overall architecture of the sedimentary bodies. The morphology of these cliffs and the position of the small catchments that reach the shoreline exert a secondary local control on the sedimentation processes and thereby on the facies development and location. These factors result in a complex architecture with large lateral variability (Figure 6). The architecture of the aeolian sedimentary bodies observed is interpreted as parabolic dunes and sand sheets. Palaeosols and aeolian deposits (dunes) cover the bedrock. These dunes were formed under strong wind conditions and the sand supply derived primarily from the exposed shelf. Directional data from the aeolianites, indicates that sand transport in the study area was predominantly towards the northwest (Figure 6), due to topographic influence. Also during cold climate intervals vegetation was less extensive [43,44], and dunes could move easily.

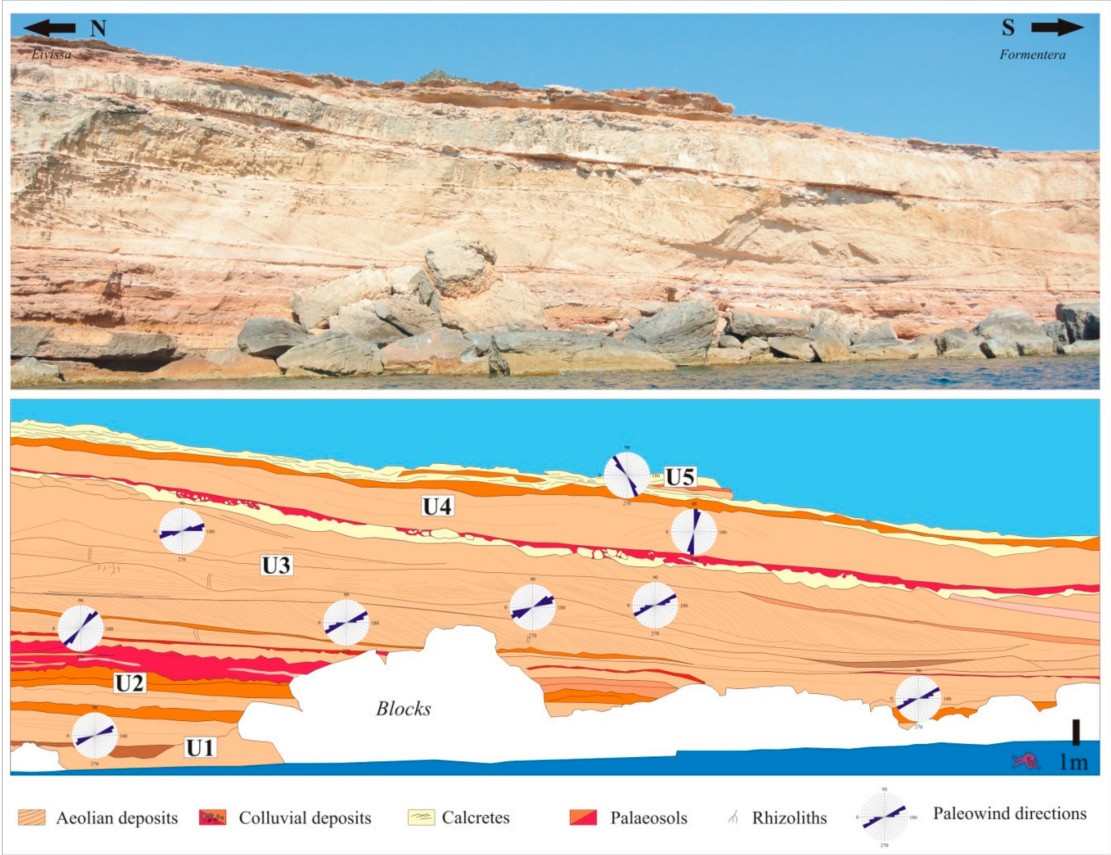

**Figure 6.** Synthetic stratigraphic sketch of the Pleistocene deposits architecture. Rose diagrams represent the main lamina dipping direction of the aeolian levels. From the base to top, comprising colluvial deposits covered by aeolian deposits (*Sht*) are observed. Unit U2, is composed by sandsheets (*Sel*) and small parabolic dunes (*Shu*) separated by silty (*Pmn*) and sandy (*Par*) palaeosols, unit U3 is composed of the different levels of parabolic dunes and sand-sheets with interdunar palaeosols (*Pmn* and *Par*). The unit U4 is composed of colluvial deposits (*Bmn*) and parabolic dunes on top. Finally, unit U5 is composed of small parabolic dunes (*Shu*) and sand-sheets (*Sel*) separated by sandy palaeosols.

The sedimentological observations allow for defined aeolian episodes and reconstruction of the landscape history of the Espalmador islet during the Pleistocene.

Based on the main erosive surfaces and palaeosols, five unconformity-bounded units are observed and indicate five aeolian sedimentation phases. These aeolian deposits represent the thickest and most

continuous layers in the study area. They show a lateral shift of sedimentary facies due the bedrock palaeotopography. Figure 4 presents a sketch of the units and in Figure 7 a 3D model for the landscape reconstruction of the aeolian accumulation on Espalmador islet.

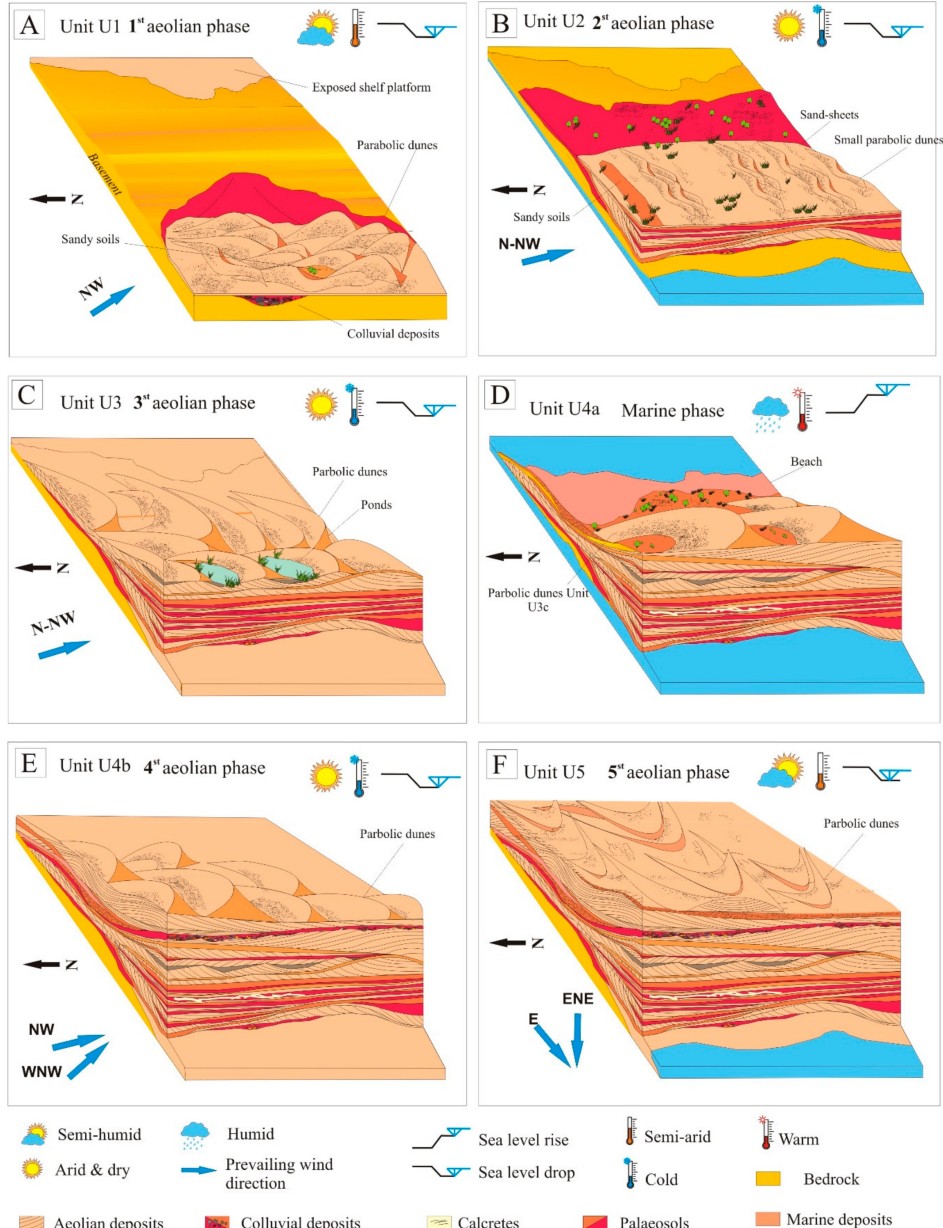

**Figure 7.** Depositional environment evolution model showing the succession of the main sedimentary environments and measured palaeocurrent directions of the coast of Espalmador islet during the Pleistocene. (**A**) First aeolian phase, formed during a low sea level, and semi arid environmental conditions. (**B**) Marine transgression during which sand ramps and coluvials were formed along cold and arid conditions. (**C**) Pronounced marine regression during an arid and dry environmental conditions, forming a coastal dune field of parabolic dunes was formed with interdunar small ponds. (**D**) Marine transgression, one meter higher of the current sea level formation of beach deposits during a humid and warm environmental conditions. (**E**) Formation of the fourth aeolian phase during a low sea level, arid and dry conditions and low temperatures, predominant wind direction of the NW-WNW. (**F**) Formation of the fifth Aeolian phase during semiarid environmental conditions, change of the predominant wind direction from NW to E-ENE.

The lower unit U1 is composed of aeolian facies (*Sht*) interbedded by silty palaeosol (*Pmn*) layers (Figure 3 Col ME 17 and) resting on the Miocene breccia bedrock. This palaeosol contains levels of subangular and heterometric clasts, originating from the bedrock. The aeolian facies level represents a high parabolic dune field of 1 to 2 m in thickness (*Sht*), which shows lateral changes (being thicker seawards and thinner inland). These aeolian sediments are separated by palaeosols representative of warmer and relatively wetter climatic conditions. Silty-clayey palaeosols, which in some cases have a high degree of bioturbation, nodules and reddish coloration due to rubefaction, indicate an increase in precipitation [45]. Otherwise, the presence of calcretes and pisolith horizons indicate events of strong aridity [44]. The succession architecture shows a dune field with active migration associated with NW paleowind direction and represent the first aeolian phase deposition. Locally, this aeolian deposit is overlaid by a silty paleosol.

Unit U2 can be divided in two sub-units (U2a and U2b). U2a is integrated by aeolian facies (*Shu* and *Sel*) the clayey very cemented palaeosol (*Pmn*) and sandy palaeosol (*Par*) ( Figure 3 Col ME 16 and Figure 7B). Sub-unit U2b is composed of the aeolian facies (*Sel*) and of different types of palaeosols (Figure 3 Col ME 17). In the lower part of this unit calcrete levels are interbedded by the clay cemented palaeosol. The aeolian levels are represented by sand-sheets (*Sel*) and small parabolic dunes (*Shu*) show laterally, difference being thicker seawards and on the southern part, and thinner inland and in the northern part of the island. The foresets show S to SE dip direction, indicating that dunes migrated in a dominant N and NW wind direction. These deposits suggest an alternation of very arid and dry phases, with a lower sea level than the current sea level., and very humid and stable periods represented by the palaeosols.

The unit U3 is separated from unit U2 by a sandy palaeosol (Figure 7C). Facies architecture and succession patterns of the deposits show three clear aeolian sub-units separated by palaeosols (U3a, U3b and U3c). The sub-unit U3a (Figure 3 Col ME 10) is composed of the aeolian facies (*Sht*) and a palaeosol (*Par*) while sub-unit U3b is composed of facies (*Sht*) and two palaeosols (*Par*) and (*Pmn*) (Figure 3 Col ME 7). Sub-unit U3c is composed of facies (*Sht* and *Sel*) and a sandy palaeosol (*Par*) (Figure 3 Col ME 6). Between the overlying dunes there are lens-like shaped pedogenetic levels interpreted as small interdune ponds. The aeolian facies represent field dunes composed of small-superimposed parabolic dunes and an interdunar palaeolagoon (*Pmn*). The paleocurrent analysis from unit U3a provides an aeolian transport to WNW and SE. The foresets of unit U3b show S to SE dip direction, indicating that dune migrated in the dominant N and NW wind direction. Unit U3c shows N to NW dip direction, also indicating that the dune field migrated in the dominant S to SE wind direction. This unit corresponds to the third aeolian deposition phase and reflects the complex field of superimposed parabolic dunes that suggest a long arid period interrupted by short humid periods forming palaeosols.

Unit U4 is separated from the unit U3 by an erosive unconformity (*Bmn*) interpreted as a colluvial deposit that contains calcretes (Figure 3 Col ME 11 and Figure 7D). Resting on this level there is a unit of aeolian facies represented by high-superimposed parabolic dunes migrating inland (ESE) and covered by a sandy palaeosol 0.5 to 1 m thickness with abundance insect trace fossils. In the northern section, concretely in cala Bocs(Col 4) (Figure 1D), and the southern section (Col 23 and 24), the morphology of the unconformity resembles a trangressive surface developed during a period of higher sea level, represented by the facies (*Sef*) that corresponds to a beach deposit (Figures 5 and 7E). The marine fossils found in the beach deposits by Butzer and Cuerda (1962) [34] and in our field work indicate warm climate conditions associated with sea level rising. This fauna speciesindicates a very littoral facies. The presence of *Littorina* species, is abundant (a small gastropod that lives on the water surface and even above the sea level, on rocky coasts), while there are no species characteristic of the Senegalese (warm) fauna found in the sediments [35]. The deposits can nevertheless be assigned to the end of Marine Isotopic Stage (MIS) 5, considering its lithology and elevation. According to Butzer and Cuerda, (1962) [34], Nolan (1895) [33] indicated the presence of Quaternary beaches with *Strombus* (*Persistrombus coronatus*) on the Espalmador islet; this has not been founded again in the Pleistocene

marine sediments of the islet, a fact that originally led to believe that the author has referred to the presence of such beaches in generic terms.

The elevation of the marine Pleistocene deposits correlates with those found in Mallorca and Menorca [35], confirming the similitudes of MIS 5 with the present.

Unit U5 is made up two aeolian levels, facies (*Shu* and *Sel*), colluvial facies and a sandy palaeosols (*Par*) with insect trace fossils (*Rebuffoichnuss casamiquelai*) ( Figure 3 Col ME 18 and Figure 7F). Aeolian facies levels resting on an erosive contact overlie colluvial facies present at the base. Dune foreset bedding show an averaged ENE dip direction. This unit is interpreted assmall parabolic dunes and sand sheet that are interbedded with sandy palaeosols. The colluvial deposits on the bottom indicates and wetter and warm period, associated with a rise sea level. The aeolian facies indicates an arid environmental change to a more arid and dry conditions with some stable periods represented by the sandy palaeosols. These Pleistocene deposits are partially covered by Holocene materials in the eastern part of the islet.

## 6. Conclusions

Based on the sedimentology and stratigraphic analysis, four sedimentary facies and two palaeosol layers are identified, which comprise five unconformity units. This analysis of the spatial distribution of the deposits and the erosion surfaces or contacts, demonstrated five periods of aeolian accumulation (parabolic dunes and sand-sheets) composed by marine sediments. The nature of the sediments implies a drop in sea level and the exposure of a high amount of marine sediments on the platform or exposed shelf areas. Also, the absence of a high amount of rhizoconcretions corresponds to cold climatic intervals with limited vegetation cover. The fact that the bedrock is totally covered by these Pleistocene deposits and their subaquatic continuity points out that the marine level was lower than the current one at the time of its formation. This probably confirms the connection between the Illes Pitiüses by an extensive dune field (composed by parabolic dunes) during a sea level low stand.

Moreover, two periods of continental deposition interpreted as colluvial deposits are observed. The formation of colluvial deposits as well as the formation of soils, local episodes of ponds seems to be related to wetter conditions. Also, the reddish color of the edaphic levels, which indicate a rubefaction process, points to a formation during humid and hotter periods. The formation of calcretes probably corresponds to arid but still warm conditions. These are interbedded with the soils, which indicates an alternation of short humid and arid episodes. The beach deposits indicate an ancient sea level one meter above the current sea level. This marine level is related lithologically, stratigraphically and faunistically with some interval of the isotopic stage MIS 5.

Finally, the succession of the aeolian, marine and colluvial deposits is interpreted as the result of a complex interaction of depositional processes generated by the Pleistocene climatic variability.

Based on the analysis presented here we suggest that wind direction and its interaction with the coastal relief as well as the fluctuations of the sea level (i.e., exposed shelf platform) were the major controls on Pleistocene coastal landscape evolution on the Espalmador islet.

**Author Contributions:** L.d.V.V.; fieldwork, writing the manuscript, performing sedimentological and composition analysis and making the figures. J.J.F., funding, performing the composition analysis and review the manuscript and A.T.-G., review the manuscript.

**Funding:** Partially of the first author capital and by the research grant MINECO CGL 2016-79246-P (AEI/FEDER, UE).

**Acknowledgments:** We thank Janina Bösken for improving the English version of the paper for her comments and suggestions. And we also want to thanks to the Govern de les Illes Balears, especially to the Conselleria de Medi Ambient, Agricultura I pesca, to accept the access and the samples collection in the Ses Salines Natural Park, also to the Ajuntament de Sant Josep de saTalaia, for the lodging in sa Casilla, especially to Räul Luna. This work is partially funded by the research grant MINECO CGL2016-79246-P (AEI/FEDER, UE).

**Conflicts of Interest:** In accordance with JMSE and the ethical obligation as a researches, we reporting that no potential conflict of interest. Part of the financing was by the capital of the first author, and partially funded by the research grant MINECO CGL 2016-79246-P (AEI/FEDER, UE). In addition, the first author is part of a PhD program at the University of Babes-Bolyai, Faculty of Environmental Sciences.

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
