# Peer review of "Geomorphological Processes and Environmental Interpretation at Espalmador islet (Western Mediterranean)"

_jmse, doi:10.3390/jmse7050144_

Round 1

Reviewer 1 Report

The manuscript provides an interesting reconstruction of the late Quaternary morpho-sedimentary evolution of a small islet of the Balearic archipelago. 3-D reconstruction of the depositional processes of the coastal landscape is largely based on classical facies analysis. The methodological approach is consolidated and the results of the paper are well presented. Thus, the paper is a good contribution that can provide a valuable contribution to the reconstruction of the recent stratigraphic evolution of the study area. I have only the following minor concerns: i) in my opinion, the title is a bit vague and should be changed in order to increase the scientific appeal of the paper; ii) the reference list should be updated according to the journal rule; iii) the first part of the introduction section is not focused and needs to be improved. In fact, it introduces only a general description of the Quaternary eustatic cycles and their influence on coastal evolution. A fuller explanation of the information gap that this paper wants to fill could improve this section of the manuscript; iii) some more details about the age of the deposits should be added.

Other minor suggestions and corrections have been added to the attached PDF-file.

Author Response

Dear Reviewer,

This document makes a serious effort to respond to all of your recommendations. I hope I have achieved it. 

Many thanks for yur efforts and recomendations.

Laura

Reviewer 2 Report

This paper presents an interesting paleogeographical reconstruction of the coastal landscape of an island in the western Mediterranean based on sedimentological and stratigraphic considerations. While very interesting from a regional perspective, I have two major issues that need to be substantively addressed before I would recommend publication:

(a) what is the scientific problem being addressed here; i.e. why are these types of regional paleogeographic reconstructions interesting and important for coastal scientists worldwide to read about? 

(b) the paper lacks placement in its context with respect to other studies of this type; what is new and different here about the methodology or insights that have not been achieved in other studies? I am having trouble ascertaining where is the novelty and/or innovation of this paper that makes it compelling reading.

Author Response

(a)    what is the scientific problem being addressed here; i.e. why are these types of regional paleogeographic reconstructions interesting and important for coastal scientists worldwide to read about?

Few papers deal with the Pleistocene of the Pitiusas Islands and even less of their numerous Islets. The study of the islets is important to look for local differences, and then to try correlating in a regional, Mediterranean and global scale. The study of the islets gives us direct and indirect information of the geomorphological processes that occurred during the Pleistocene related to sea level fluctuations in the Mediterranean. In this way this study wants to understand the present and the future evolution of the system, as well as to obtain models that would provide tools for an integrated and sustainable management of the coastal area(with special attention to the Mediterranean)in the context of the present-day and future climatic change.

It is important to stand out the relationship of cold events with aeolianite deposition in this central part of the western Mediterranean.

(b) the paper lacks placement in its context with respect to other studies of this type; what is new and different here about the methodology or insights that have not been achieved in other studies? I am having trouble ascertaining where is the novelty and/or innovation of this paper that makes it compelling reading.

The Pleistocene deposits developed mainly in the coastal zone allow to obtain a good paleoclimatic and geomorphological information from the Pliocene to the present, that are coherent and can be contrasted with the data obtained in other parts of the planet. They are also indicators of the sea level, that are fundamental to understand and predict the environmental changes which will affect in a more or less distant future.This interest comes because an important part of the population tends to settle in these coastal areas.In addition, the Balearic Islands has based its socioeconomic development on the use of the coast. It should be mentioned that in the current context of climate change discussion, the understanding of the climate and the consequent environmental conditions of the past, should allow to infer the future evolution of the climate and, therefore, the evolution of the future coastal zone, without role that human activity has been able to play on it. Therefore Espalmador is a good area for study, located in the middle of the two major islands of the Pitiusas, with an important Pleistocene sequence and continues along the whole island, with the alternation of aeolian, colluvial, marine deposits and the formation of paleosols, provides an extensive and good amount of palaeoclimatic information, paleoenvironmental and fluctuations of the sea level, which would help to better understand the processes occurred in the past and understand the present, as well as build a future climate scenario in this area of the western Mediterranean

Reviewer 3 Report

General comments:

The manuscript by Del Valle et al. titled ‘Description and environmental interpretation of the Pleistocene coastal dune fields at s’Esplamador islet (Western Mediterranean) presents an interesting study integrating lithostratigraphic logging, grain-size analyses and X-ray diffraction measurements on 24 sediment columns distributed over the s’Esplamador islet. Main aim of the work is to distinguish different sedimentological facies and interpret these deposits in terms of the triggering environmental conditions. The results of the work reveal seven sediment facies comprising five main depositional units connected with varying climate and sea level conditions. I support publication of this paper in Journal of Marine Science and Engineering. However, before publication, some points should be clarified, described and discussed in more detail.

Specific comments:

Methods:

The method section of the manuscript can be improved to unambiguously support the results and interpretations of the work. For example, more details about the lithostratigraphic logging method would be helpful. How were the lithological descriptions and measurements performed (this is not clear for me working with sediment cores)? At which scales (mm-m?) are the analyses conducted? How were sub-samples for e.g. gain-size measurements extracted? What size do these samples have? How many samples were extracted per unit, to receive significant/repeatable results?

In addition, more details about the grain-size measurements could be helpful. How does the IMAGEJ software determine grain-sizes from photographs? How many repeated measurements were performed per unit to receive a significant (mean) value? What are the differences of the individual grain-size measurements in one unit?

Finally, it is not specified how the HCl dissolution (M, time, temperature?) helps with the determination of the carbonate content.

Interpretation of the sediment facies/depositional units:

The description/interpretation of the work would benefit from information and a map describing the spatial distribution and orientation of the investigated 24 s’Esplamador sediment columns (maybe as an inset in Fig. 3 and a table indicating coordinates, length of columns,…). In addition, a connection of this spatial distribution with the presented order of the sediment columns in Fig. 3 would help to easier follow the discussion about the depositional processes. For example, it could help to group the sediment columns in Fig. 3 in N-S or W-E profiles, or something similar.

Mentioning more often the differences between individual columns (with numbers), like in line 232 (and a reference to Fig. 3), would provide the reader substantial (available) information about prevailing wind directions.

Moreover, a reference supporting the inferred depositional mechanism of the individual sediment facies would enhance confidence in the interpretations.

Finally, a more tabulared version of Fig. 4, would allow to easier assess the features of the seven sediment facies, compared to the now presented text.

Figure captions:

The figure captions would benefit from more information about e.g. the location and names of the sediment columns, to be understandable without a careful reading of the text. Symbols within the figures are not defined. See the ‘detailed comments’ below.

English:

Please check English wording throughout the manuscript. Check for missing and additional spaces. See a non-exhaustive list of suggestions in the ‘detailed comments’ below.

Detailed comments:

Consistently use ‘s’Esplamador’ or ‘Esplamador’.

Lines 24-30: Add ‘for example’ before ’marine deposits. There are much more quaternary environmental archives than mentioned in the list.

Line 34: What does ‘general’ mean?

Lines 35-38: Be more specific about the ‘quality’ of the deposits on the Baleares.

Line 49: Delete the text before the comma.

Line 60: Replace ’comprised’ with ‘composed’.

Lines 62-64: Please rephrase that sentence.

Line 65: Replace ‘citation’ with e.g. ‘description’.

Lines 65-66: One sentence paragraphs are not consistent with English grammar.

Line 93: Add ‘measurements’ between ‘size’ and ‘was’.

Line 104: ‘Figs’ not ‘Fig’.

Line 107: Provide a reference for the given color code.

Line 107: Use singular for the facies description. Facies Sht is…

Line 137: Add an interpretation and a reference for the ‘shallow marine facies’, like for the other facies.

Line 145: Delete ‘In general terms’.

Line 187: Add a space between ‘a’ and ‘3’.

Fig. 1: Indicate the meaning of the stars in C and D. I cannot see the 24 investigated columns in Fig 1D (see my specific comment regarding the spatial distribution of the sediment columns). Add A-D to the figure.

Fig. 2: Note the orientations of the photographs. Add a scale. Add the location of the photographed costal formations in Fig. 1D.

Fig. 3: Add names of profiles (from-to). Add a unit for the y axis. The x-axis is not readable. Enlarge the hardly readable text, particularly in the legend. Based on which specification are the columns ordered from left to right (see specific comment above).

Fig. 6: Add scales. Add the definition of U1 to U5 in the figure caption.

Fig. 7: Do the arrows reflect wind directions? What does the Sun and thermometer symbols stand for? I do not understand the symbols to the right of the Thermometers.

Author Response

Dear Reviewer, 

I have made and incorporated all the necessary changes to improve the paper who suggested me.

Thank you for your contribution and efforts.

Laura

Round 2

Reviewer 2 Report

Paper is OK with the revisions made. It is made more clear what is the global significance of the work.

Reviewer 3 Report

Thank you for the comprehensive revision of the manuscript. Please increase the text size in Figure 3, it is still very difficult/impossible to read, and add a unit to the y-axis.